# Does Rebound Pain after Peripheral Nerve Block for Orthopedic Surgery Impact Postoperative Analgesia and Opioid Consumption? A Narrative Review

**DOI:** 10.3390/ijerph16183257

**Published:** 2019-09-05

**Authors:** Olufunke Dada, Alicia Gonzalez Zacarias, Corinna Ongaigui, Marco Echeverria-Villalobos, Michael Kushelev, Sergio D. Bergese, Kenneth Moran

**Affiliations:** 1Department of Anesthesiology, The Ohio State University Wexner Medical Center, 520 Doan Hall, 410 West 10th Avenue, Columbus, OH 43210, USA (O.D.) (C.O.) (M.E.-V.) (M.K.) (K.M.); 2Department of Anesthesiology, Stony Brook University, Stony Brook, New York, NY 11794, USA

**Keywords:** rebound pain, hyperalgesia, peripheral nerve blocks, orthopedic surgeries

## Abstract

Regional anesthesia has been considered a great tool for maximizing post-operative pain control while minimizing opioid consumption. Post-operative rebound pain, characterized by hyperalgesia after the peripheral nerve block, can however diminish or negate the overall benefit of this modality due to a counter-productive increase in opioid consumption once the block wears off. We reviewed published literature describing pathophysiology and occurrence of rebound pain after peripheral nerve blocks in patients undergoing orthopedic procedures. A search of relevant keywords was performed using PubMed, EMBASE, and Web of Science. Twenty-eight articles (n = 28) were included in our review. Perioperative considerations for peripheral nerve blocks and other alternatives used for postoperative pain management in patients undergoing orthopedic surgeries were discussed. Multimodal strategies including preemptive analgesia before the block wears off, intra-articular or intravenous anti-inflammatory medications, and use of adjuvants in nerve block solutions may reduce the burden of rebound pain. Additionally, patient education regarding the possibility of rebound pain is paramount to ensure appropriate use of prescribed pre-emptive analgesics and establish appropriate expectations of minimized opioid requirements. Understanding the impact of rebound pain and strategies to prevent it is integral to effective utilization of regional anesthesia to reduce negative consequences associated with long-term opioid consumption.

## 1. Introduction

While regional anesthesia is considered a great tool for reducing postoperative pain and opioid consumption, rebound pain following nerve blocks may reduce or even negate its overall benefits. Rebound pain is a condition characterized by hyperalgesia after the peripheral nerve block wears off. While patients experience less pain and lower opioid consumption when the peripheral nerve block is functioning, the occurrence of rebound pain often leads to a counter-productive increase in opioid consumption thereafter.

Postoperative pain is one of the most feared surgical complications reported by patients, particularly for orthopedic surgeries, which are frequently followed by a painful recovery [1,2,3]. Among strategies to prevent acute pain from progressing to chronic pain, opioid-based therapy has become a mainstay of intraoperative and postoperative pain management [4,5,6]. However, in spite of their efficacy in the management of acute and chronic pain, opioids are strongly associated with unwanted effects [7]. Most significantly, opioids are the leading cause of drug overdose deaths and for a good number of patients their first exposure to opioids will be in the perioperative period [8,9,10]. The surge in opioid related deaths has prompted an urgent need to expedite efforts towards a drastic reduction of perioperative opioid use [10,11].

The use of regional anesthesia as part of multimodal analgesia strategies has grown exponentially in order to maximize pain control, decrease opioid requirements, and promote early mobility and rehabilitation in the perioperative period in patients undergoing orthopedic surgeries [12,13,14,15,16]. Nevertheless, peripheral nerve blocks have been associated with rebound pain, a state of hyperalgesia with an onset between 8 and 24 h after block administration [17,18,19]. According to Williams et al., rebound pain is the ‘quantifiable difference in pain scores when the block is working versus the increase in acute pain encountered during the first few hours after the effects of peri-neural single-injection or continuous infusion local anesthetics resolve’ [20]. In a study comparing patients who had single shot brachial plexus blocks with those who had general anesthesia without block for wrist fracture surgery, the authors reported a 40% incidence of severe postoperative pain in patients who had single shot brachial plexus blocks versus 10% in those who had general anesthesia. Additionally, unplanned physician visits in the first 48 h due to pain was greater in the brachial plexus group when compared to patients who received general anesthesia [21]. Thus, the beneficial effects harnessed in terms of effective pain management and reduced opioid consumption when the block is functioning may be compromised by the occurrence of rebound pain [22]. Rebound pain is still a poorly understood concept and few studies have evaluated its full impact on the use of regional anesthesia as a strategy to reduce long-term pain and opioid consumption.

We reviewed published literature describing the pathophysiology of rebound pain and reporting its occurrence after peripheral nerve blocks for orthopedic procedures.

## 2. Methods

A search for manuscripts published in English between January 2014 and January 2019 was performed for this review using PubMed, EMBASE and Web of Science. The keywords included “Rebound pain”, “rebound hyperalgesia”, “rebound pain and regional anesthesia”, “rebound pain and orthopedic surgery”, “rebound pain and continuous block”, “rebound pain and single shot block”, “rebound pain and peripheral nerve block”, “rebound postoperative pain”, “rebound pain and regional analgesia”, “recurrent postoperative pain and peripheral nerve block”, and “recurrent postoperative pain and regional anesthesia”. Randomized controlled trials, case reports, clinical trials, comparative and observational studies, meta-analyses, systematic reviews, and review articles describing the use of peripheral nerve block (e.g., single shot or with catheter insertion) either alone or combined with general anesthesia were included in our review. In contrast, pre-conference abstracts, articles unrelated to rebound pain in the context of regional anesthesia for orthopedic surgeries, and trials involving non-orthopedic surgical populations were excluded.

## 3. Results

A total of 768 manuscripts were identified from which 408 were duplicates. Therefore, 360 articles underwent title/abstract screening. Following this, 332 articles were excluded due to being unrelated to rebound pain in orthopedic surgeries (*n* = 325), non-orthopedic surgical populations (*n* = 2), study protocols (*n* = 3), and pre-conference abstracts (*n* = 2). Hence, a total of 28 articles were included in this review. Half of these manuscripts involved patients who experienced rebound pain after upper limb blocks, whereas the other half involved patients with rebound pain following lower limb blocks.

The upper limb blocks discussed were mainly interscalene, suprascapular, axillary and infra-clavicular nerve blocks for procedures such as shoulder arthroplasty, shoulder arthroscopy, distal radius fracture fixation, and wrist fracture fixation. On the other hand, femoral, sciatic, lumbar plexus, and fascia iliacus blocks, as well as periarticular injections were used for lower limb procedures including total knee replacement (majority), arthroscopic hip surgeries, ankle fracture surgeries, and anterior cruciate ligament reconstruction.

Single shot and continuous infusion blocks were similarly used among trials. Local anesthetics (i.e., bupivacaine, ropivacaine, levobupivacaine, liposomal bupivacaine, and lidocaine) were administered alone, in combination (e.g., bupivacaine with lidocaine), or with adjuvants such as corticosteroids (triamcinolone), dexmedetomidine, buprenorphine, clonidine, and dexamethasone [12,17,23].

In addition, the majority of the blocks were instituted pre-emptively for the purpose of analgesia (as part of the multimodal approach) combined with sedation or general anesthesia. All the studies reported rebound pain to varying degrees.

## 4. Discussion

### 4.1. Pathophysiologic Hypothesis for Rebound Pain

#### 4.1.1. Abnormal Specific Nerve Fibers Spontaneous Hyperactivity

Preclinical studies in rats using a sciatic nerve block with ropivacaine 0.5% showed that 3 h after the mechanical analgesia resolved the animals presented transient heat hyperalgesia, although there was no evidence of persistent changes in mechanical sensitivity or nerve damage [24]. Similarities of rebound pain with neuropathic pain (burning pain, hyperalgesia) suggest that abnormal spontaneous C-fiber hyperactivity and nociceptor hyper-excitability without mechanical nerve lesion may play a role as one of the potential pathophysiologic mechanisms [24,25,26]. Kleggetveit et al. examined nociceptor’s function using microneurography in patients with burning neuropathic pain and demonstrated that patients presenting pain showed hypersensitization and hyper-excitability of mechano-insensitive C-nociceptors. Additionally, the study showed an axonal involvement in the sensitization process [26].

#### 4.1.2. Patient-Related Factors

Patient-related risk factors that play a role in the incidence and severity of rebound pain after peripheral nerve blocks have also been reported. Severe pre-operative pain, age less than 60 years old, female sex, and psychosocial factors such as catastrophic perception of pain and depression have been identified as the most important factors influencing this complication of peripheral nerve blocks [27,28,29,30,31]. There is evidence showing a lower incidence and severity of rebound pain in patients >65 years old [31,32]. A recent prospective, observational study by Sort et al. in patients undergoing emergency surgery for ankle fracture reported that rebound pain was less intense in patients >60 years of age, whilst patients between 20 and 60 years old reported higher levels of rebound pain [32]. This difference in frequency and severity of rebound pain in elderly patients can be explained by some proven factors such as changes in pain perception and deep tissue nociception [30]. Lautenbacher et al. in a prospective study conducted in young and elderly volunteers demonstrated a decrease in pressure pain threshold with age, whereas heat pain threshold remained unaffected and threshold for non-noxious stimuli became higher in elderly individuals [30].

#### 4.1.3. Surgery-Related Factors

Surgical injury can produce an abnormal level of plasticity at the peripheral nociceptor level and in the central neurons involved in receiving and processing the direct and indirect inputs [33,34]. Damage to the peripheral nociceptors provokes a continuous firing of pain signals leading to either an exaggerated response to normally painful stimuli (hyperalgesia) or a noxious response to normally non-painful stimulation (allodynia) [35,36]. The state of central pain sensitization retains and amplifies the nociceptive signals, which has been called ‘pain memory’ [37]. When blockade of the signal transduction subsides, the sensation of pain becomes amplified [36,37]. This mechanism of pain memory may explain why increasing the duration of the peripheral block with a continuous technique and/or the use of adjuvants may not always produce a significant decrease in the occurrence and severity of rebound pain [20].

#### 4.1.4. Anesthesia-Related Factors

Reversible neural toxicity of local anesthetics has been proposed as a possible contributing mechanism for the occurrence of rebound pain [38]. Local anesthetics reduce perioperative pain and the surgical response to pain by reversibly blocking voltage-gated sodium channels (VGSC) and the axonic propagation of nerve signals [39]. Another beneficial effect of local anesthetics is an anti-inflammatory action mediated by the time-dependent and reversible inhibition of the G protein-coupled receptor, which has been reported in vitro and in vivo [40]. Other preclinical studies showed systemic and local anti-inflammatory effects in different animal models [41,42]. Some human studies have demonstrated a systemic anti-inflammatory effect of lidocaine in different painful clinical situations [43,44], as well as in surgical interventions [45]. Local anesthetics also target potassium channels and N-methyl-D-aspartate (NMDA) receptors [39]. Conversely, other studies have shown that local anesthetics have pro-inflammatory activity. Gordon et al. in a prospective study in patients undergoing dental surgery found that bupivacaine stimulated higher cyclo-oxygenase 2 (COX-2) gene expression, increased prostaglandin E_2_ (PGE_2_) production, and a greater level of pain after termination of local anesthetic (LA) effect [46].

The nerve injury related to a regional block is a combination of several etiopathogenic mechanisms such as direct mechanical nerve lesion, neurotoxicity of the LA, ischemia, or a combination of these [47,48]. Insertion of a needle into the nerve during a peripheral nerve block (PNB) does not typically result in significant injury unless intraneural injection occurs, which can cause elevated intraneural pressure that exceeds capillary occlusion pressure and subsequent nerve ischemia and focal demyelination [47,48]. Ischemia can also result from direct damage or occlusion of the epineural vessels, as well as perineural hemorrhage [48]. High volume, increased concentration, and prolonged exposure to the LA reduce neural blood flow [49]. Inter-fascicular injection of the solution and high-pressure injection may disrupt the inter-fascicular vasculature and produce nerve ischemia [48]. It is not fully understood whether nerve injury during PNB worsens rebound pain after the block wears off.

Different types of LA may produce neurotoxicity through different mechanisms [50,51]. Amide and ester local anesthetics significantly stimulate DNA fragmentation, endoplasmic reticulum calcium depletion, intra-mitochondrial calcium overload, and decline in membrane potential and uncoupling of oxidative phosphorylation with critical reduction in ATP synthesis [48,50,52]. Research evidence shows that bupivacaine activates overproduction of reactive oxygen species (ROS) and autophagy [53,54]. The sum of the aforementioned intracellular events leads to damage and apoptosis of Schwann cells, macrophage infiltration, and myelin damage [52,55,56].

### 4.2. Rebound Pain and Specific Nerve Blocks

#### 4.2.1. Rebound Pain and Interscalene Nerve Block

Shoulder surgeries, including arthroscopic procedures, have been associated with moderate to severe pain [57]. The growing trend for performing shoulder surgery in the ambulatory setting while maintaining standards for pain control, expediting post anesthesia recovery time, and achieving rehabilitation goals further exemplifies the demand for optimal pain management in these patients.

Interscalene brachial plexus block (ISBPB), the most widely used peripheral nerve block for shoulder surgery, has been found to be efficacious and cost effective [58,59]. While the short-term analgesic benefits of single-shot ISBPB have been elucidated, the resolution of the block typically occurs within the first 24 h. A meta-analysis by Abdallah et al. demonstrated pain control at 8 h at rest and an opioid sparing effect up to 24 h [60]. Rebound pain may occur 8 to 24 h after block institution irrespective of the choice of local anesthetic, volume, or concentration [18,60,61,62]. Kim et al. compared the incidence of rebound pain in patients undergoing arthroscopic shoulder surgery performed under single shot block versus continuous patient-controlled interscalene block. Patients in the continuous infusion block group experienced an improved analgesic profile 12 h postoperatively (*p* < 0.001) and did not report rebound pain in the first 24 h [17]. While continuous perineural local anesthetic infusions may offer an option for decreasing rebound pain, there are significant limitations. Shoulder surgeries, such as rotator cuff repairs, are frequently associated with greater than 7 days of acute postoperative pain outlasting the duration of commonly utilized perineural local anesthetic infusions [63]. Additionally, prolonged phrenic nerve palsy, inadvertent catheter removal, and limitations in resources for placement and subsequent follow up further limit broad range continuous catheter techniques [64,65].

Another study attempting to examine rebound pain was performed by Namdari et al. comparing interscalene brachial plexus block (ISBPB) using 0.5% ropivacaine with periarticular local anesthetic (LA) infiltration using liposomal bupivacaine. While ISBPB significantly decreased pain scores up to eight postoperative hours (*p* < 0.001), there was no difference at 16 h (*p* = 0.348), and pain scores were paradoxically lower in the periarticular infiltration group at 24 h (*p* = 0.021). The authors report that patients in the ISBPB experienced rebound pain as seen by an increase in visual analog scale (VAS) pain scores from 1.4 (at 8 h) to 4.9 (at 24 h). Patients in the periarticular infiltration group did not have a significant change in pain scores over the 24-h postoperative period. [66]. However, the use of different local anesthetics in the groups made it difficult to determine if rebound pain non-occurrence in the periarticular LA group was a result of the avoidance of ISBPB or the use of liposomal bupivacaine. It was interesting to note that the total morphine equivalent consumption (intraoperative plus postoperative) was still lower in the ISBPB group due to the significantly lower use in the intraoperative period (*p* < 0.001) [66].

#### 4.2.2. Rebound Pain and Suprascapular Nerve Block (SSNB)

The use of SSNB has been described either solely or in combination with axillary nerve block (ANB) to provide analgesia for arthroscopic shoulder surgery. When comparing SSNB used alone versus combined SSNB + ANB, the authors found improved mean VAS in the first 24 h, better patient satisfaction, and fewer episodes of rebound pain in the combination group [67]. Similarly, another study comparing three groups, namely IV PCA alone, IV PCA with SSNB, and IV PCA with SSNB + ANB, found the combination of SSNB + ANB with PCA to provide superior analgesia up to 12 h, with all 3 groups experiencing rebound pain at 12 and 36 h postoperatively [68]. A study comparing the efficacy of combined SSNB + ANB with ISBPB demonstrated that while pain management was better in the first 8 h in the ISBPB group, the combined SSNB + ANB provided better pain management thereafter with no short- or long-term rebound pain [62]. Similar to findings observed for periarticular infiltration, SSNB + ANB, while providing less initial analgesia as compared to ISBPB, may lead to overall lower pain scores and opioid consumption.

In one particularly notable study, Lee et al. conducted a randomized controlled trial involving 48 patients undergoing arthroscopic rotator cuff repair. Twenty-four patients had ultrasound guided interscalene block with arthroscopically guided suprascapular block while the other 24 patients had interscalene block alone using ropivacaine. The authors reported lower VAS scores for up to 28 h post-operatively and better patient satisfaction for up to 36 h in the combination group. All patients in both groups had rebound pain. Additionally, in the ISBPB + SSNB group, six patients experienced “double rebound” phenomenon, in which they had two distinct periods of hyperalgesia corresponding to the expected termination of effect of each individual block. There was however a statistically significant delay in onset (*p* < 0.001) and a significantly reduced severity (p = 0.001) of rebound pain in the ISBPB + SSNB group compared with ISBPB alone [69]. The duration of ISBPB and SSNB has been reported as 8 h and up to 24 h respectively [62,70]. Hence, it appears that a longer (differential) duration of block provided by the combination of these blocks is a more important factor than either of the blocks alone in minimizing the impact, even if not the occurrence of rebound pain. It may therefore be possible to harness this difference in their durations of action to maximize analgesic effects in single shot blocks as an alternative to continuous perineural infusions.

#### 4.2.3. Rebound Pain and Other Brachial Plexus Blocks

Analgesia for surgical procedures below the clavicle can be provided by supraclavicular, infraclavicular, or axillary approaches to the brachial plexus [71]. Galos et al. compared general anesthesia using intravenous fentanyl analgesia (*n* = 18) with infraclavicular brachial plexus block (without general anesthesia) using Lidocaine with epinephrine plus bupivacaine (*n* = 18) in patients undergoing distal radial fracture fixation. The brachial plexus group experienced significantly lower pain scores and required less morphine within the first two postoperative hours (*p* < 0.001) [72]. However, the pain scores progressively increased and rebound pain occurred at 12 and 24 h in the brachial plexus group resulting in significantly greater opioid intake in the brachial plexus block group from 6 to 12 h after surgery (<0.001) [72]. As a result, the overall opioid consumption was not different between the two groups. Ganta et al. attempted to examine rebound pain following single-shot infraclavicular block as compared to continuous infraclavicular block for patients undergoing distal radius fractures. The authors were unable to demonstrate an improvement of rebound pain in the 12–24 h postoperative period or in opioid consumption [65]. Unfortunately, research examining the potential supremacy of supraclavicular or axillary nerve blocks in improving rebound pain compared to infraclavicular nerve blocks is limited.

#### 4.2.4. Rebound Pain and Femoral Nerve Blocks

Rebound pain has also been reported in lower limb nerve blocks [14,15,19,73]. Femoral nerve blocks provide effective analgesia for knee surgeries with limitations being rebound pain and increased risks of falls due to quadriceps weakness [19,74,75]. Xing et al. described superior pain control with femoral nerve block when compared to general anesthesia alone in the first 6 h post-operatively [19]. However, rebound pain prevented any significant reduction in post-operative opioid consumption in comparison with the general anesthesia alone group. Perhaps one of the more useful studies attempting to define and quantify rebound pain was done on patients receiving femoral nerve blocks. The authors quantified rebound pain by subtracting the lowest pain score reported in the 12 h before the block wears off from the highest pain score during the first 12 h after the resolution of the block. This interesting idea incorporates the relative nature of pain perception between patients and measures pain escalation following the end of block duration. The authors appropriately recognize that it is sometimes difficult to define the exact point of block resolution [20].

Pain outcomes following femoral nerve blocks done at the adductor canal are similar to conventional femoral nerve blocks in patients undergoing arthroscopically assisted anterior cruciate ligament repair under general anesthesia [74]. While rebound pain was not mentioned for either group, the adductor canal block group had better pain control and less opioid consumption in the first four hours post-operatively (*p* = 02). However, there was no significant difference in overall pain and morphine consumption for the four day period of observation after surgery. These findings are consistent with the other findings discussed in this article that suggest that the improved pain control in the early postoperative period, as provided by a nerve block, does not necessarily translate to an overall decrease in postoperative opioid consumption.

Youm et al. compared VAS scores in patients who received a femoral nerve block (FNB) with those who received a peri-articular injection (PAI) and those who had a combination of both [76]. Improved analgesia was noticed with a PAI in the first eight hours postoperatively compared to FNB, likely due to the involvement of pain in the sciatic distribution. Interestingly, their findings suggest that the addition of an FNB to a PAI prevented the occurrence of rebound pain seen with a PAI alone. Stathellis et al., on the other hand, found that patients who received continuous femoral nerve block (CFNB) had rebound pain (after discontinuation of the catheter/infusion), when compared to the postoperative intra-articular infusion (PIAC) group [15]. Perhaps the addition of adjuvants, particularly ketorolac with its anti-inflammatory property and anti-nociceptive effect, contributed to the non-occurrence of rebound pain in the PIAC group. These types of studies can be difficult to interpret due to a lack of standardization of local anesthetic dosing, concentration, and the addition of various adjuvants when comparing nerve block or pain management approaches.

#### 4.2.5. Rebound Pain and Combined Sciatic/Saphenous Nerve Blocks

Henningsen et al. interviewed patients who received combined sciatic and saphenous nerve blocks performed as the primary anesthetic for ankle surgeries. In spite of a prescription for preemptive acetaminophen and ibuprofen, some patients experienced postoperative rebound pain lasting up to two hours and in some cases not relieved by the rescue analgesia, morphine. Even though the potential for rebound pain and need for preemptive analgesia was explained to the patients before the block was instituted, most of the patients were still unsure when to use medications, especially when it seemed one block wore off before the next. This may have contributed to increased pain with associated increased opioid requirements. The authors highlighted the important concept of good patient instruction and communication about utilizing preemptive analgesic medication to prevent pain once one the block has worn off [77].

#### 4.2.6. Rebound Pain in Lower Extremity Nerve Blocks When Compared to Neuraxial Blocks

While the focus of this article is rebound pain following peripheral nerve blocks, central neuraxial blocks are established anesthetic techniques for lower limb surgeries and offer a valuable comparison to peripheral blocks when evaluating rebound pain. It is known that a quicker onset of anesthesia can be expected with spinal blocks while the femoral-popliteal block has a longer duration of post-operative analgesia [78]. Future studies comparing spinal anesthesia with peripheral nerve blocks may show more insight into the advantages and drawbacks of each and the effect of the occurrence of pain and/or rebound pain on overall total opioid consumption in both groups [31].

Wolff et al. reported significantly lower pain scores up to 2 h post-operatively in patients undergoing arthroscopic hip surgery under lumbar plexus versus fascia iliaca block [73]. Regardless of pain scores, most of the patients received a standard dose of oxycodone as soon as they could tolerate oral medications. Although these patients did not have a significant reduction in opioid consumption or decreased time to discharge from the post anesthesia care unit, it is interesting to note the lack of occurrence of postoperative rebound pain in either group. Perhaps this was as a result of the pre-emptive analgesic effect of the opioid medication.

### 4.3. Adjuvants to Local Anesthetics for Peripheral Nerve Blocks and Rebound Pain

Adjuvants have been combined with local anesthetic agents including clonidine, dexmedetomidine, dexamethasone, buprenorphine, midazolam, epinephrine, tramadol, magnesium, morphine, and others [23,79]. These adjuvants, in addition to prolonging the duration of analgesia, help to reduce overall dose requirements for local anesthetics [23]. However, concerns for potential neural side effects and toxicity of these adjuvants exist [79]. Knight et al. found, contrarily, that dexmedetomidine when used as adjuvant appears to improve analgesia without increasing the risk of local anesthetic neurotoxicity [23]. Likewise, in a mouse model, perineural dexamethasone when added to bupivacaine appeared to prevent the axon degeneration and demyelination that was seen in mice that had bupivacaine alone for sciatic nerve block [38]. Rebound hyperalgesia was noticed in the bupivacaine group alone but not in the bupivacaine + dexamethasone (high and low dose) groups [38]. This suggests that the effect of local anesthetics on the nerves resulting in rebound pain is attenuated by the addition of perineural dexamethasone. This effect of dexamethasone was, however, not noticed when it was administered systemically. Dexamethasone also prolonged the anti-nociceptive effect of bupivacaine in a dose dependent manner [38]. Dexamethasone as well as buprenorphine, clonidine, dexmedetomidine, and magnesium appear to show the most consistent ability, in literature, to prolong the duration of peripheral nerve blocks [79]. Higher doses of perineural dexamethasone have been associated with higher rebound pain compared with lower doses (or even perhaps a specific dose of 2 mg for plexus blocks), while for buprenorphine the converse appears to be the case [80]. Hence, the optimal use of these adjuvants to harness their beneficial effects in the reduction and possibly prevention of rebound pain needs to be further explored.

## 5. Conclusions

Rebound pain after peripheral nerve blocks for orthopedic surgeries may reduce the overall benefits of regional anesthesia. Multimodal strategies such as preemptive opioid analgesia before the block wears off, the utilization of intra-articular or intravenous steroidal and non-steroidal anti-inflammatory drugs, the use of adjuvants in nerve block solutions to lengthen the period of analgesia, and continuous blocks (versus single shot blocks) may be utilized to reduce the occurrence of rebound pain.

A future area of study that may provide insight into the nature of rebound pain is whether a complete absence of pain during a nerve block results in a lower pain tolerance once it wears off. Conversely, the question should be asked whether less dense blocks that permit some pain breakthrough are perceived as having a relatively less intense rebound pain experience. Similarly, research on local anesthetic concentration may also reveal differences in rebound pain experiences.

Different institutional protocols have been proposed in patients undergoing orthopedic surgery as part of a multimodal strategy incorporating peripheral nerve blocks. However, an optimal approach to reducing rebound pain remains uncertain. A greater understanding of the impact of rebound pain on long-term pain control and opioid consumption and the development of strategies to prevent it will be required before we can fully determine whether regional anesthesia can be used as an effective tool for reducing long-term opioid use and abuse.

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
