# Peer review of "Does Rebound Pain after Peripheral Nerve Block for Orthopedic Surgery Impact Postoperative Analgesia and Opioid Consumption? A Narrative Review"

_ijerph, 2019, doi:10.3390/ijerph16183257_

Round 1
Reviewer 1 Report
It is an interesting and valuable review paper. Here are my concerns and suggestions:
The angle should be changed. You fixed yourself on decreasing opioid consumption and found in the review process that it is not that important. The most important is the optimal analgesia in the period after the local anesthesia wears off. In the light of the above, the title should be changed. Besides, it is misleading: what does "its" refer to?
Line 46:
1. That is unjustified simplification: 1. people die of drug overdose. 2. opioids are the leading cause of it. 3. A good number of patients start with opioids in the perioperartive period. 2. Please give reference to the statement: a good number of patients their first exposure to opioids will be in the perioperative period. Line 129 Please consider skipping brand names. Lines 152, 178, 182, and further "narcotic" is not a medical term; please use "opioid"instead Line 159 'double rebound' - please, add a sentence with clarification to the uninitiated readers Line 208 An explication for this TLA required Lines 245-248 You found an outstanding (and perhaps the paramount) solution for the prevention of rebound pain, and you did not mention in your conclusions! Line 257 I am not sure if references 48-49 apply to the topic of this study. Lines 272-275 This reference does not refer to rebound pain but the pain in general. Line 306 An explication required Lines 252-353 It should be discussed earlier and not appear in the conclusion section. Lines 359-362 Completely redundant and unjustified conclusion.
Author Response
Reviewer #1
Q1: The angle should be changed. You fixed yourself on decreasing opioid consumption and found in the review process that it is not that important. The most important is the optimal analgesia in the period after the local anesthesia wears off. In the light of the above, the title should be changed. Besides, it is misleading: what does "its" refer to?
A1: Taking in consideration your suggestions about the title, we have changed the title of our manuscript to “Does Rebound Pain after Peripheral Nerve Block for Orthopedic Surgery Impact Post-operative analgesia and Opioid Consumption? A Narrative Review.”
Q2: Line 46: 1. That is unjustified simplification: 1. people die of drug overdose. 2. opioids are the leading cause of it. 3. A good number of patients start with opioids in the perioperative period. Please give reference to the statement: a good number of patients their first exposure to opioids will be in the perioperative period.
A2: Query addressed accordingly, references added
Q3: Line 129: Please consider skipping brand names
A3: brand name removed as per query suggestion
Q4: Lines 152, 178, 182, and further "narcotic" is not a medical term; please use "opioid" instead
A4: Narcotic terms removed as per query suggestion
Q5: Line 159 'double rebound' - please, add a sentence with clarification to the uninitiated readers
A5: A “double rebound” explanation was added as per query suggestion
Q6: Line 208 an explication for this TLA required
A6: Query addressed accordingly
Q7: Lines 245-248 you found an outstanding (and perhaps the paramount) solution for the prevention of rebound pain, and you did not mention in your conclusions
A7: This topic was addressed in our conclusion. The word opioid was added to clarify the sentence “Multimodal strategies such as preemptive opioid analgesia before the block wears off…”
Q8: Line 257 I am not sure if references 48-49 apply to the topic of this study
A8: The references above described the pathophysiology of rebound pain, which is similar found in neuropathy pain
Q9: Lines 272-275 This reference does not refer to rebound pain but the pain in general
A9: The statement and reference (Sort et al) prior to the one referenced ( Lautenbacher et al) in the comment discussed that patient’s over 60 have less rebound pain than those under 60. The reference mentioned in this comment (Lautenbacher) was then used to demonstrate evidence that supports that patients older than 60y/o have higher threshold for pain which may explain the findings referred to previously about rebound being less severe in those >60 – see Sort, R.; Brorson, S.; Gögenur, I.; Nielsen, J.K.; Møller, A.M. Rebound pain following peripheral nerve block anaesthesia in acute ankle fracture surgery: An exploratory pilot study. Acta Anaesthesiologica Scandinavica 2019, 63, 396-402.
Q10: Line 306 An explication required
A10: Query resolved as per suggestion
Q11: Lines 252-353 It should be discussed earlier and not appear in the conclusion section
A11: Query resolved as per suggestion
Q12: Lines 359-362 Completely redundant and unjustified conclusion
A12: Thank you for your feedback on this section. In this paragraph, the authors are suggesting future areas of research and are not attempting to make any conclusions. Anecdotally, patients who have zero pain before a nerve block wears off seem to be much more surprised by the sudden return of pain. Conversely, patients who have some breakthrough pain do not seem to find rebound pain to be as startling. This may, in part, be because they take some pain medicine for the small amount of pain they do experience or could be because of the emotional/psychological effects of pain (prepared vs unprepared response). Because we could not find studies that review this concept and consider it clinically anecdotal we did not interject the concept into the article. However, we would like to suggest that it may be interesting to look at in the future as an area of potential research that can shed light on optimizing nerve blocks to reduce overall opioid consumption.
Reviewer 2 Report
see attachment.

Author Response
Reviewer #2
Q1: Environmental Research and Public Health does not appear to be an appropriate journal for this article. This is a medical manuscript of interest to surgeons and anesthetists who would not access this journal for the information
A1: This review article was written as an invitation for the International Journal of Environmental Research and Public Health (special issue on the effectiveness and safety of high dose opioid therapy). The authors feel that regional anesthesia can play a role in improving the safety of opioid use and reducing the need for high dose opioid therapy by providing an effective transition to controlled opioid utilization following surgery. However, as the article addresses, rebound pain can negate these potential benefits. We feel that information on rebound pain and its prevention can influence the discussion on patient safety in the context of opioid therapy.
Q2: Line 215….the comment regarding addition of ketorolac is important as this highlights the importance of a multimodal approach, especially with ketorolac known for its potent analgesic affect. This could be emphasized more, in general, by adding references to support this. Acetaminophen is not appropriate. From experience, opioids alone post-operatively, are not effective in managing orthopedic pain.
A2: Query addressed per suggestion. Added: anti-nociceptive effects. This statement regarding Ketorolac was added as an opinion of the authors based on the literature findings by Stathellis et al.
Q3: Hyperalgesia not hyper-algesia throughout.
A3: Query addressed per suggestion
Q4: Line 58 ………who had general anesthesia: add - without block
A4: Added –without block, as per query suggestion
Q5: Line 110… rehabilitation goals: add, after goals
A5: Query addressed per suggestion
Round 2
Reviewer 1 Report
Query 10 has remained unsolved despite the Author's assertion.
Query 12 - the word "abuse" is redundant. The Authors seem to be struggling against drug abuse, not to try to find the optimal way for analgesia or long-term opioid therapy.